# 3D Chitin Scaffolds from the Marine Demosponge *Aplysina archeri* as a Support for Laccase Immobilization and Its Use in the Removal of Pharmaceuticals

**DOI:** 10.3390/biom10040646

**Published:** 2020-04-22

**Authors:** Jakub Zdarta, Tomasz Machałowski, Oliwia Degórska, Karolina Bachosz, Andriy Fursov, Hermann Ehrlich, Viatcheslav N. Ivanenko, Teofil Jesionowski

**Affiliations:** 1Faculty of Chemical Technology, Institute of Chemical Technology and Engineering, Poznan University of Technology, Berdychowo 4, 60965 Poznan, Poland; tomasz.g.machalowski@doctorate.put.poznan.pl (T.M.); oliwia.degorska@gmail.com (O.D.); karolinabachosz@gmail.com (K.B.); 2Institute of Electronics and Sensor Materials, TU Bergakademie Freiberg, Gustav-Zeuner str. 3, 09599 Freiberg, Germany; andriyfur@gmail.com (A.F.); hermann.ehrlich@esm.tu-freiberg.de (H.E.); 3Wielkopolska Center for Advanced Technologies (WCAT), Poznan University str. 10, 61614 Poznan, Poland; 4Department of Invertebrate Zoology, Biological Faculty, Lomonosov Moscow State University, 119992 Moscow, Russia; ivanenko.slava@gmail.com

**Keywords:** chitin scaffolds, marine scaffolds, enzyme immobilization, laccase, tetracycline, pollutant removal

## Abstract

For the first time, 3D chitin scaffolds from the marine demosponge *Aplysina archeri* were used for adsorption and immobilization of laccase from *Trametes versicolor*. The resulting chitin–enzyme biocatalytic systems were applied in the removal of tetracycline. Effective enzyme immobilization was confirmed by scanning electron microscopy. Immobilization yield and kinetic parameters were investigated in detail, in addition to the activity of the enzyme after immobilization. The designed systems were further used for the removal of tetracycline under various process conditions. Optimum process conditions, enabling total removal of tetracycline from solutions at concentrations up to 1 mg/L, were found to be pH 5, temperature between 25 and 35 °C, and 1 h process duration. Due to the protective effect of the chitinous scaffolds and stabilization of the enzyme by multipoint attachment, the storage stability and thermal stability of the immobilized biomolecules were significantly improved as compared to the free enzyme. The produced biocatalytic systems also exhibited good reusability, as after 10 repeated uses they removed over 90% of tetracycline from solution. Finally, the immobilized laccase was used in a packed bed reactor for continuous removal of tetracycline, and enabled the removal of over 80% of the antibiotic after 24 h of continuous use.

## 1. Introduction

Traditionally, diverse polymer-, carbon-, inorganic-, and hybrid-based ultralight porous three-dimensional (3D) materials have been used for enzyme immobilization [1,2,3,4]. Recently, modern porous 3D printing scaffolds with complex internal structures and channels have been the focus of much attention [5]. Our scaffolding strategy [6] is based, however, on naturally prefabricated 3D scaffolds of poriferan (sponge) origin. These unique biopolymer-containing constructs offer alternative immobilization matrices that can be isolated from demosponges cultivated worldwide, to provide appropriate supports for a broad range of enzymes. Thus, demosponges of the order Dictioceratida (also known as commercial bath sponges) [7,8] represent a renewable source of proteinaceous spongin scaffolds which have recently been reported as effective in applications in extreme biomimetics [9,10,11,12], waste treatment [13,14,15,16,17], electrochemistry [18], and enzyme immobilization [19]. Meanwhile, marine demosponges of the order Verongiida have been recognized as a renewable source of uniquely pre-structured 3D chitinous scaffolds [20,21,22,23,24,25,26,27,28] which have found applications in tissue engineering [6,21,29,30,31,32,33,34], drug release [35], the development of hybrid materials [36,37,38,39,40], and environmental science [41,42]. Chitin of invertebrate origin has previously been studied by researchers in protein immobilization as a matrix for the immobilization of enzymes [43,44,45,46,47,48], including lipases [49,50] and papain [51]. However, in this study we investigated the ability of a unique, ready-to-use 3D chitinous scaffold isolated from the marine demosponge *Aplysina archeri* (Figure 1) to immobilize laccase as a selected enzyme, and an innovative application of this unique system in the removal of tetracycline by simultaneous adsorption and catalytic conversion.

The laccase (EC 1.10.3.2) is an enzyme classified as multicopper oxidoreductase, that occurs in numerous higher plants, several bacteria, and is secreted by many fungi [52]. Laccase catalyzes the oxygen reduction reaction directly to water with the simultaneous oxidation of polyphenols, aminophenols, polyamines, and lignins. The active center of the laccase contains four adjacent copper atoms, which represent three types distinguished by their specific properties. Type I copper gives the enzyme molecule blue color and is the site of oxidation of the substrate. Copper type II and two atoms of copper type III form a three-atomic assembly in which binding and reduction of molecular oxygen to water occurs. During each catalytic cycle of laccase, there is a reduction of one oxygen molecule to two water molecules, accompanying oxidation of four substrate molecules to four substrate radicals [53]. Laccases find a variety of applications in many industries due to a number of reactions involving them. This enzyme is used to remove impurities in wastewater thanks to the phenol oxidation reaction and to degrade industrial waste [54]. It is also used on a large scale in the process of lignin degradation and detoxification of aromas generated in this process [55]. In order to increase its activity and easier separation of the enzyme from the post-reaction mixture laccase can be immobilized on a variety of carriers, such as chitinous sponges.

## 2. Materials and Methods

### 2.1. Chemicals and Materials

Samples of the verongiid demosponge *Aplysina archeri* (Higgin, 1875) (Figure 1) were collected at depths of 10–25 m by scuba divers around the Caribbean islands of Saint Vincent and Curaçao in May–June 2017, during the Pacotilles expedition. All permits required for the study were obtained during that expedition, which complied with all relevant regulations [26]. The reagents used for isolation of chitin sponge skeleton—sodium hydroxide, acetic acid, and hydrogen peroxide—were supplied by Merck (Darmstadt, Germany). Reagents for enzyme immobilization—laccase from *Trametes versicolor* (EC 1.10.3.2, activity ≥0.5 U/mg), 50 mM phosphate buffer (at pH 7–9), and 100 mM acetate buffer (at pH 3–6)—were supplied by Sigma Aldrich (St. Louis, MO, USA). Tetracycline (>99%), 2,2′-azino-bis(3-ethylbenzothiazoline-6-sulfonic acid) (ABTS) and Bradford reagent used for examination of the quantity of immobilized enzyme were supplied by Sigma Aldrich (St. Louis, MO, USA).

### 2.2. Isolation of Chitin Sponge Skeletons

#### Modified Standard Method

The isolation of chitinous scaffold from *A. archeri* was performed by a modified version of the standard method proposed by Ehrlich et al. in 2007 [20]. First, a selected fragment of the sponge (see Figure 1) was placed in deionized water for 1 h at 24 °C to remove water-soluble compounds. A deproteinization process was then carried out using 2.5 M NaOH (Th. Geyer GmbH & Co. KG, Renningen, Germany) at 37 °C for 24 h. The skeletal scaffold was next washed with distilled water to neutral pH, and treated with 20% acetic acid (Th. Geyer GmbH & Co. KG, Renningen, Germany) at 24 °C for 6 h to remove residual calcium carbonates. It was then neutralized by rinsing with deionized water [26]. Deproteinization and demineralization of the sponge skeleton were repeated for 144 h to obtain a completely soft and pigment-free chitinous scaffold (Figure 2) for further use as a support in enzyme immobilization.

### 2.3. Laccase Immobilization

Prior to laccase immobilization, the chitinous scaffolds (Figure 2C) were cut into pieces weighing 5 mg. Then, 5 mg of the isolated scaffold was placed in a vial, to which 2 mL of laccase solution in acetate buffer at pH 5 and concentration 5 mg/mL was added. The mixture was placed in an incubator (IKA Werke GmbH, Staufen im Breisgau, Germany) and was shaken at 150 rpm for 1 h at 25 °C. After enzyme immobilization the samples were removed from the mixture, washed with acetate buffer at pH 5 to remove unbound enzyme, and used in tests involving the removal of tetracycline. The supernatant after immobilization was subjected to spectrophotometric measurements to evaluate the quantity of immobilized enzyme and the immobilization yield.

### 2.4. Characterization of Immobilized Enzymes

The amount of the immobilized enzyme was calculated based on the spectrophotometric measurements using the Bradford protein assay method [56]. The amount of immobilized enzyme, expressed in mg/g, was determined as the difference between the initial amount of enzyme and the final laccase concentration in the mixture after immobilization, relative to the mass of the chitin scaffold. Immobilization yield (%) was calculated by considering the difference in the amounts of the enzyme before and after immobilization in both, supernatant after immobilization and acetate buffer used to remove unbound enzyme, and the volume of the solution used in this process.

Activity assays for free and immobilized laccase were performed spectrophotometrically, based on a model reaction using 2,2′-azino-bis(3-ethylbenzothiazoline-6-sulfonic acid) (ABTS) as a substrate. Briefly, 10 mg of free or immobilized enzyme was added to 5 mL of a mixture containing 10 mM of ABTS in phosphate buffer at pH 5. The reaction was carried out for 60 min at 25 °C. After the process, spectrophotometric measurements were made at wavelength 420 nm. One unit of free or immobilized laccase activity was defined as the amount of enzyme needed to convert 1 mM of ABTS per minute under the assay conditions. Based on the results, using a standard calibration curve for ABTS, the specific activity of the free and immobilized enzyme (U/mg) was calculated as the initial enzyme activity retained, respectively, per unit mass of enzyme and per unit mass of enzyme and support. The activity retention (%) of immobilized laccase is presented as the percentage activity of the immobilized laccase, relative to the catalytic activity of the free enzyme.

The kinetic parameters of the free and immobilized laccase—the Michaelis–Menten constant (K_M_) and the maximum reaction rate (V_max_)—were examined based on the above-mentioned ABTS oxidation reaction using solutions of substrate at concentrations ranging from 0.01 to 10 mM, performed under optimal assay conditions. The apparent kinetic parameters (K_M_ and V_max_) of free and immobilized laccase were calculated using Hanes–Woolf plot.

The storage stability of free and immobilized laccase was tested spectrophotometrically over 30 days of storage at 4 °C in acetate buffer at pH 5. Storage stability was determined using ABTS as a substrate, according to the methodology described above, by evaluation of the activity retention of free and immobilized enzyme at specified time intervals.

The thermochemical stability of both free and immobilized laccase over time was examined after incubating the samples for 120 min under optimum pH and temperature conditions (pH 5, temperature 25 °C). Spectrophotometric measurements were performed based on the reaction using ABTS as a substrate. The relative activity of free and immobilized enzyme was determined at specified time intervals. For clearer presentation of the data, in these experiments the initial activity of free laccase was defined as 100% activity. The inactivation curves of free and immobilized enzyme, the inactivation constant (k_D_) and the half-life (t_1/2_) were calculated based on the linear regression slope.

### 2.5. Removal of Tetracycline

The main objective of the study was to use the obtained biocatalytic systems for efficient removal of tetracycline in various pH and temperature conditions and using antibiotic solutions at various concentrations. To determine the contributions of adsorption and catalytic conversion to total tetracycline removal, experiments were performed to evaluate the efficiency of adsorption of tetracycline by pure chitin sponge skeleton and by the sponge skeleton with immobilized enzyme following thermal inactivation, as well as the efficiency of removal of the antibiotic by simultaneous adsorption and catalytic conversion by the produced biocatalytic systems.

#### 2.5.1. Adsorption of Tetracycline by Pure Chitin Scaffolds

Before the overall efficiency of tetracycline removal was evaluated, a determination was made of the efficiency of adsorption of the antibiotic by the pure chitin scaffolds. For this purpose, 10 mg of the chitinous scaffold was placed in vials, to which 10 mL of the tetracycline solution at appropriate concentration was added. The process was carried out for 1 h. To examine the effect of pH, buffer solution was used to adjust the pH to values ranging from 3 to 9 (acetate buffer at pH 3–6 and phosphate buffer at pH 7–9). The experiments were performed at 25 °C, using tetracycline solution at concentration 1 mg/L. Tests to establish the effect of temperature on the adsorption efficiency were performed at temperatures ranging from 5 to 65 °C (10 °C step) using tetracycline solution at pH 5 and concentration 1 mg/L. The effect of antibiotic concentration on the efficiency of its adsorption was determined at 25 °C using solutions at pH 5 and concentrations 0.1, 0.5, 1.0, and 3.0 mg/L. After the adsorption process the chitinous sponges were separated and the reaction mixture underwent further spectrophotometric measurements.

#### 2.5.2. Adsorption of Tetracycline by Chitin Scaffolds with Thermally Inactivated Enzyme

In the next step, a determination was made of the efficiency of adsorption of tetracycline by chitin scaffold with immobilized enzyme following thermal inactivation. For this purpose, the products obtained after the immobilization stage were placed in a dryer for 2 h at 80 °C for enzyme inactivation, and then used in the adsorption experiment. The adsorption tests were carried out for 1 h using 10 mL of the tetracycline solution, to which 10 mg of the chitinous scaffold with inactive enzyme was added. The effect of pH was examined over pH ranging from 3 to 9, at 25 °C using tetracycline solution at concentration 1 mg/L. The effect of temperature on the efficiency of adsorption of tetracycline was examined at temperatures from 5 to 65 °C (10 °C step) using antibiotic solution at concentration 1 mg/L and acetate buffer at pH 5. The effect of concentration of tetracycline solution on the efficiency of its adsorption was examined at 25 °C using solutions at pH 5 and concentrations 0.1, 0.5, 1.0, and 3.0 mg/L. After adsorption the chitinous skeletons were separated and the reaction mixture underwent spectrophotometric measurements.

#### 2.5.3. Removal of Tetracycline by Simultaneous Adsorption and Catalytic Conversion

The next step involved determination of the efficiency of removal of tetracycline by simultaneous adsorption and catalytic conversion, catalyzed by the systems with immobilized enzyme. For this purpose, to 10 mg of the freshly obtained biocatalytic systems, 10 mL of tetracycline solution at the appropriate concentration was added. The process was carried out for 1 h. To evaluate the effect of tetracycline concentration on the rate of its removal, solutions at concentrations 0.1, 0.5, 1.0, and 3.0 mg/L, in acetate buffer at pH 5, were used. The process was performed at 25 °C. To study the effect of pH, solutions at concentration 1 mg/L at pH ranging from 3 to 9 were tested at 25 °C. The effect of temperature on the efficiency of removal of tetracycline was examined over the temperature range 5–65 °C (10 °C step) using antibiotic solution at concentration 1 mg/L at pH 5. After the removal experiments the biocatalytic systems were separated and the reaction mixture underwent spectrophotometric measurements.

#### 2.5.4. Reusability of Immobilized Laccase

The reusability of the immobilized laccase was examined according to the methodology described above, by measuring the efficiency of removal of tetracycline over 10 repeated biocatalytic cycles using 10 mL of tetracycline solution at pH 5 and concentration 1 mg/L at 25 °C. After each removal cycle the chitinous scaffolds with immobilized laccase were separated from the reaction mixture, washed with pH 5 acetate buffer to remove unreacted substrates and products from the chitin scaffold, and transferred to fresh tetracycline solution.

### 2.6. Removal of Tetracycline in Packed Bed Reactor

Continuous removal of tetracycline by the chitinous scaffolds with immobilized laccase was performed in a packed bed bioreactor containing 500 mg of chitinous scaffold with immobilized laccase (the amount of immobilized enzyme was approx. 1 g). A schematic illustration of the packed bed reactor used in the study is presented in Figure 3. The process was performed for 24 h at temperature 25 °C using tetracycline solution at concentration 1 mg/L at pH 5. The tetracycline solution was placed in a substrate vessel and was passed through the column at a flow rate of 2 mL/min using a peristaltic pump. The effluents from the column were collected in separate vessels at specified intervals and underwent spectrophotometric measurements. The efficiency of removal of tetracycline was calculated using the following equation (Equation (1)):(1)$$\frac{Adsorption}{Removal}\;Efficiency\;\left(\%\right)=\frac{C_{i}-C_{t}}{C_{i}}$$
where *C_i_*. denotes the initial tetracycline concentration and *C_t_*. denotes the final tetracycline concentration after treatment.

### 2.7. Analytic Techniques

The morphology of the obtained chitinous materials before and after immobilization was examined by transmission electron microscopy (TEM) performed on a Hitachi HT7700 instrument (Hitachi, Japan) working in high contrast (HC) mode and operating at 100 kV. Prior to measurements, an appropriate quantity of the sample was dispersed in 2 mL of deionized water with the use of ultrasounds (Ultrasonic bath, Cavotator, Anaheim, CA, USA) and then 5 µL of the solution was applied on the nickel mesh grid covered with a carbon film. SEM photographs were obtained using an EVO40 scanning electron microscope (SEM, Zeiss, Oberkochen, Germany).

The structural features of the obtained chitinous scaffolds were observed using an advanced imaging and measurement system consisting of a Keyence VHX-6000 (Keyence, Tokyo, Japan) digital optical microscope and the swing-head zoom lenses VH-Z20R (magnification up to 200×) and VH-Z100UR (magnification up to 1000×).

The quantity of immobilized enzyme, immobilization yield, activity of free and immobilized enzyme, and efficiency of removal of tetracycline were examined based on spectrophotometric measurements using a Jasco V750 UV-Vis spectrophotometer (Jasco, Tokyo, Japan). The measurements were performed at 595, 420, and 355 nm, respectively, for the Bradford method, ABTS, and tetracycline removal experiments. The final concentrations of laccase after the immobilization process, ABTS after oxidation, and tetracycline after treatment were obtained using calibration curves for bovine serum albumin, ABTS, and tetracycline, respectively. The rate of adsorption/removal of tetracycline (%) in all of the aforementioned experiments was calculated based on the Equation (1).

### 2.8. Statistical Analysis

All experiments and measurements in this study were performed in triplicate, and error values are defined as means ± standard deviation. Statistically significant differences were determined by one-way ANOVA using Tukey’s test, performed in SigmaPlot 12 (Systat Software Inc., Los Angeles, CA, USA). Statistical significance was established at a level of *p* < 0.05.

## 3. Results

### 3.1. Characterization of Products after Immobilization

#### 3.1.1. Morphological Analysis

In the first step of characterization, the surface morphology of the chitinous scaffolds before and after immobilization of laccase was determined based on analysis of SEM and TEM images (Figure 4).

From Figure 2 and Figure 4a it can be seen that the surface of chitinous scaffold from *A. archeri* before enzyme deposition is relatively uniform and smooth, being typically slightly folded after drying. Furthermore, no kind of microdamage was observed in the SEM and TEM images. By contrast, upon laccase immobilization, numerous aggregates with irregular shape and a size of around 500 nm are present on the surface of the chitinous scaffold. These aggregates may be interpreted as agglomerates of the enzyme molecules, and their presence confirms effective enzyme immobilization.

#### 3.1.2. Characterization of the Immobilization Process and Kinetic Parameters of the Free and Immobilized Laccase

After confirmation of efficient immobilization, the process as well as the resulting chitin-based biocatalytic systems were characterized in terms of the process yield and the activity of the immobilized enzymes (Table 1). Furthermore, apparent kinetic parameters (K_M_ and V_max_) of free and immobilized laccase were determined to examine changes in the substrate affinity of the enzyme upon immobilization.

From Table 1 it can be seen that after 1 h of laccase immobilization from solution at concentration 5 mg/L, the relatively high immobilization yield of 91% was achieved. This results in an extremely high quantity of laccase immobilized on the chitinous scaffolds (around 1800 mg of the enzyme was immobilized per 1 g of the support). It is also evident from the data that almost 90% activity was retained by the immobilized laccase, as its specific activity was 52.5 U/mg, compared with 56 U/mg for the free enzyme. The study also included evaluation of the kinetic parameters of the free and immobilized laccase. Upon immobilization a slight increase in the value of the Michaelis–Menten constant (K_M_), from 0.093 to 0.113 mM, was recorded. These results indicate a slightly lower substrate affinity in the case of the immobilized biomolecules, reflected in the lower values of the maximum reaction rate (0.048 mM/s) obtained for the immobilized laccase as compared to the free enzyme. Consequently, a lower catalytic efficiency (0.516 1/s) was also recorded for the immobilized laccase. Although the data show a slightly lower substrate affinity in the case of laccase bonded with the chitinous scaffold, the changes do not exceed 20%, clearly indicating that the obtained systems can be considered as a robust biocatalyst for further applications.

#### 3.1.3. Storage Stability and Thermochemical Stability of Free and Immobilized Enzymes

The next part of the study concerned evaluation of changes in the relative activity of the free and immobilized laccase over time under storage and under process conditions (pH 5, temperature 25 °C) in order to determine their storage stability and thermochemical stability. The results are presented in Figure 5.

From Figure 5a it can be seen that the activity of the free laccase decreased slightly from the first day of the storage stability test, and after 15 days reached around 80%. After that time a more gradual decrease in relative activity was recorded, and after 30 days of storage the relative activity of the free laccase did not exceed 50%. By contrast, the relative activity of the immobilized enzyme remained unaltered over the first 5 days of storage, and then began to decrease slightly. After 15 days it reached 93%, and after 30 days of storage at 4 °C almost 85% of the initial catalytic activity was retained, which is over 50% higher than the value for free laccase. The investigation also included determination of thermochemical stability profiles of the free and immobilized laccase under process conditions (pH 5, temperature 25 °C). The results (Figure 5b,c) clearly show that, although the initial activity of the immobilized laccase was slightly lower (see Table 1), the enzyme bound to the chitinous skeletons displayed a significant improvement in thermochemical stability as compared to the free biomolecules. After 30 and 120 min of incubation at process conditions, the immobilized laccase retained, respectively, 83% and 77% relative activity, indicating that the chitin-based biocatalytic systems were quite stable and prevented inactivation of the enzyme. Meanwhile, the activity of free laccase was gradually reduced: after incubation in the same conditions, after 15 and 120 min it retained, respectively, around 80% and less than 40% of its activity. Furthermore, the inactivation constant (k_D_) and half-life (t_1/2_) of free and immobilized laccase were examined and compared based on a linear regression slope. The results are in agreement with the data presented above; the values of k_D_ and t_1/2_ for the immobilized enzyme were found to be 0.0038 1/min and 182.4 min, respectively, while for the free enzyme they were 0.0174 1/min and 39.8 min. Thus, the inactivation constant of the immobilized laccase was more than four times lower than that of the free enzyme, while the enzyme half-life was increased by 4.5 times upon immobilization. The presented data clearly indicate that chitinous sponge skeletons are suitable matrices for laccase immobilization and protect the immobilized biomolecule against thermal and chemical inactivation.

### 3.2. Removal of Tetracycline

Following confirmation of the effective immobilization of laccase and evaluation of the stability of the produced biocatalytic systems, in the next step of the investigation the immobilized laccase was applied in the removal of tetracycline from water solutions under various process conditions, including different pH, temperature, and tetracycline solution concentration. This was the main objective of the study. Furthermore, as removal of tetracycline took place by simultaneous adsorption and biocatalytic conversion, we decided to follow these processes in detail to examine their contributions to the total removal of the antibiotic. It should be additionally noted that experiments on the use of pure chitinous scaffolds and free enzyme for adsorption and conversion of tetracycline under the most suitable process conditions (pH 5 and temperature 25 °C) result in removal rates of around 80% and 60%, respectively (full data not presented), indicating that total removal of the antibiotic by the separately applied techniques is not possible to achieve.

#### 3.2.1. Effect of Concentration of Tetracycline Solution on Efficiency of Its Removal

In the first stage of the investigation, the effect of the concentration of tetracycline solution, in a range from 0.1 to 3.0 mg/L, on the efficiency of its adsorption, catalytic conversion, and total removal was determined at pH 5 and temperature 25 °C. The results are presented in Figure 6.

From Figure 6 it is evident that from solutions at concentration 0.1, 0.5, and 1.0 mg/L tetracycline was totally removed after 60 min of the process. A decrease of about 20% in the total efficiency of removal of the antibiotic was observed in the case of a solution at the highest concentration (3.0 mg/L). The data indicate that removal of tetracycline occurred as a result of simultaneous adsorption by the chitinous skeletons and catalytic conversion by the immobilized laccase. Furthermore, the higher is the tetracycline concentration, the lower the efficiency of its adsorption, which falls from 80% at concentration 0.1 mg/L to 42% for a 3.0 mg/L solution of tetracycline. By contrast, the efficiency of enzymatic conversion increased with increasing concentrations of antibiotic in the solution, reaching around 40% for a concentration of 1.0 mg/L. The exception is the solution at the highest concentration, for which the efficiency of catalytic conversion was lower (around 35%) as compared to the 1.0 mg/L solution, and for which the lowest adsorption efficiency was recorded among all of the tested samples.

#### 3.2.2. Effect of pH of Tetracycline Solution on Efficiency of Its Removal

It is recognized that pH may significantly affect both adsorption of the pollutant and the catalytic action of the immobilized laccase. Therefore, in the next step of the investigation, the effect of the pH of tetracycline solution on the efficiency of its removal was examined over a wide pH range from 3 to 9, at temperature 25 °C and using tetracycline solution at concentration 1.0 mg/L. The data obtained are presented in Figure 7.

It can be seen that the total efficiency of removal of tetracycline increased from 85% at pH 3 to reach 100% at pH 5, and then began to decrease, to around 55% at pH 9. Nevertheless, it should be noted that the total removal efficiency exceeded 90% in a pH range from 4 to 7, and remained above 80% at pH 3 and pH 7. These results are significantly higher than the efficiencies of adsorption by pure chitinous sponges and conversion by free enzyme, which were 80% and 60%, respectively, at pH 5, and decreased further on even slight changes in pH. It should be recalled that the removal of tetracycline took place by simultaneous adsorption and biocatalytic conversion. The adsorption process showed comparable efficiencies, at around 55%, at pH values ranging from 3 to 7; a decrease of around 15% in this parameter was recorded at highly basic pH (pH 8 and pH 9). By contrast, the contribution of catalytic conversion to the total removal of tetracycline is significantly higher in acidic conditions, reaching a maximum of around 40% at pH 5 and pH 6. Further increases in pH caused the efficiency of enzymatic action to fall to 30% at pH 7 and less than 15% at pH 9.

#### 3.2.3. Effect of Temperature on Efficiency of Removal of Tetracycline

Process temperature is another important parameter which may affect the efficiency of both adsorption and catalytic conversion, and as a consequence the total rate of removal of tetracycline. Therefore, the effect of temperature was examined over a wide range, from 5 to 65 °C, using tetracycline solution at pH 5 and concentration 1.0 mg/L. The results are presented in Figure 8.

From Figure 8 it can be seen that the efficiency of removal of tetracycline rose with increasing temperature, reached its maximum (100%) at 25 and 35 °C, and then decreased with further increase of the process temperature. However, although 100% removal was recorded only at 25 and 35 °C, the rate remained above 90% at temperatures of 45 and 55 °C, and over 80% of the antibiotic was removed at 15 °C and even at 65 °C. These results indicate that the obtained biocatalytic systems are capable of effectively eliminating tetracycline by simultaneous sorption and catalytic action over a much wider temperature range than in the case of adsorption by pure sponges and conversion by free laccase. Nevertheless, it should be highlighted that, although the efficiencies of adsorption and catalytic conversion exhibited similar profiles over the whole analyzed temperature range, the adsorption efficiency was about 20% higher than the efficiency of catalytic conversion of the antibiotic, reaching a maximum of around 60% at 35 °C. Notwithstanding, even at 65 °C, over 50% of the tetracycline was removed by adsorption. From Figure 8 it is also evident that the highest contribution of catalytic conversion to antibiotic removal occurred at temperatures ranging from 25 to 45 °C, and reached around 40%. At higher and lower temperatures, lower rates of catalytic conversion were recorded; however, even at 65 °C, over 25% of the pollutant was removed by the catalytic action.

#### 3.2.4. Reusability of Immobilized Laccase

Immobilization appears to be the most promising technique for improving enzyme stability, resulting in the possibility of multiple use of immobilized biomolecules in repeated reaction cycles. This property is of particular interest from an industrial point of view, since it facilitates cost reduction and enhances process control. Therefore, in the present study, the recyclability of the immobilized laccase was examined over 10 repeated batch tests of tetracycline degradation under optimal process conditions (Figure 9).

From Figure 9 it is evident that the immobilized biocatalyst offers exceptional reusability, as the efficiency of removal of tetracycline decreased only slightly over 10 reaction cycles. Furthermore, over the first three catalytic cycles the biocatalytic system is capable of removing 100% of tetracycline from the solution, and even after 10 repeated steps the immobilized laccase enables removal of over 90% of the tetracycline. Although the contributions of each technique to the total removal of the antibiotic over repeated biocatalytic cycles were not determined in this part of the study, the observed decline in removal efficiency may be related to both a decrease in the sorption efficiency and decline in the catalytic properties of the immobilized laccase.

#### 3.2.5. Removal of Tetracycline Using a Packed Bed Bioreactor

To evaluate the possible practical application of the produced chitin-based biocatalytic systems in the removal of antibiotics, such as tetracycline, from wastewaters, a packed bed bioreactor containing chitinous scaffolds with immobilized laccase was constructed for the first time for continuous removal of tetracycline. The process was performed over 24 h under the most suitable conditions for the highest removal of the antibiotic. Results are presented in Figure 10.

The experiment was performed over 24 h; however, it should be highlighted that the process time was measured from the collection of the first drop of effluent after treatment. From Figure 9 it is evident that over the first 6 h of the process total removal of the tetracycline was achieved. Further, a slight drop in the removal efficiency was recorded; however, even after 16 h of the continuous process, over 90% of the antibiotic was removed. Finally, 83% of the tetracycline was removed by the biocatalytic system after 24 h of continuous operation, by way of simultaneous adsorption and catalytic action. This provides clear indication of the enormous potential of the produced materials in practical applications.

## 4. Discussion

As was stated above, the main objective of the study was to use *A. archeri* sponge skeletons as a unique source of 3D chitinous scaffolds to serve as a support for enzyme immobilization, and to apply the obtained biocatalytic systems in the removal of tetracycline, as a model pollutant present in wastewaters. Effective immobilization of the enzyme was confirmed, and the effect of various process conditions on the efficiency of removal of the antibiotic was examined. Finally, a new packed bed bioreactor was constructed to evaluate the removal rate in a continuous process.

### 4.1. Characterization of Products Following Immobilization

In the first stage of the investigation, the morphology of the chitinous scaffolds before and after laccase immobilization was examined based on SEM and TEM images to check the suitability of the scaffolds as supports for enzyme deposition and to confirm effective binding of the enzyme. As shown in Figure 4, the chitinous scaffolds have an open, three-dimensional structure that facilitates transfer of the substrate and products to the immobilized molecules and reduces transfer limitations. The clearly visible, irregular shapes with sizes of a few micrometers, present on the chitin fibers after immobilization, should be interpreted as an enzyme aggregates. This clearly confirms effective deposition of the enzyme onto the surface of the chitinous scaffolds. Similar observations confirming laccase attachment were made in our previous study, in which scaffolds based on *Hippospongia communis* spongin were used for immobilization of laccase [19].

As shown, a very high immobilization yield of 91% was achieved, corresponding to the extraordinary amount of immobilized enzyme (1820 mg of laccase deposited per 1 g of chitinous scaffold). The high immobilization yield and large quantity of deposited enzyme may be explained by the exceptional sorption properties of *A. archeri* chitin [26] as well the presence of numerous functional groups typical for this aminopolysaccharide, mainly carbonyl and hydroxyl, that are capable of providing effective and stable enzyme binding [46]. However, it should be highlighted that although the interactions formed are stable, they are based mainly on hydrogen bonds and adsorption interactions. For this reason, interference in the structure of the enzyme and its distortion upon immobilization are limited [57]. This is a possible explanation of the high specific activity of the immobilized laccase (52.5 U/mg) as compared to the free enzyme (56 U/mg), corresponding to an activity retention of 89% [58]. By contrast, Das et al. immobilized laccase on magnetic iron nanoparticles and obtained specific activity of the immobilized protein was about 15 U/mg, that was almost two times lower as compared to free laccase (30 U/mg) [59]. Similar results were reported by Lin et al. who noticed specific activity of 20.1 and 30.1 U/mg for immobilized and free enzyme, respectively. However, in their tests chitosan/CeO_2_ microspheres were used as a support in the immobilization of laccase from *Trametes versicolor* [60]. In another study by Ramirez-Montoya et al. the specific activities of free laccase and laccase immobilized on a mesoporous carbon obtained from pecan shells were found to be 10.85 and 0.038 U/mg, respectively [61]. Therefore, it might be assumed that presented in our study significantly higher specific activity of the immobilized laccase indicates production of an effective biocatalyst.

Furthermore, kinetic parameters of the free and immobilized enzymes were determined, to investigate changes in enzyme–substrate affinity before and after immobilization. A slight increase in the K_M_ value, of about 20%, and a similar drop in the maximum rate were obtained for immobilized laccase as compared to the free biomolecule, indicating a lower substrate affinity and lower reaction rate. These results are probably related to the fact that upon immobilization some of the active sites of the immobilized laccase can be blocked, reducing their accessibility and impairing the catalytic properties of the immobilized enzyme [62]. However, it should be noted that due to the open structure of the chitinous matrix and limited deformation of the enzyme structure, only a slight worsening of catalytic properties was observed. Similar observations were made in our previous study concerning adsorption immobilization of laccase on poly(l-lactic acid)-co-poly(ε-caprolactone) electrospun nanofibers [63]. By contrast, in a study by Olajuyigbe et al. [64], laccase was immobilized by entrapment using calcium and copper alginate beads. Due to the creation of strong diffusional limitations, almost three times higher Michaelis–Menten constants were obtained. In summary, therefore, the present results clearly indicate that the obtained immobilized enzymes may be considered as effective biocatalytic systems for practical application, for instance in the removal of hazardous compounds.

Nevertheless, the retention of high catalytic properties by enzymes after immobilization is not in itself sufficient, as these systems should also provide significant improvements in thermal and chemical stability as compared to the free enzyme. In this study, both the thermochemical stability and storage stability of the free and immobilized laccase were examined and compared under optimal process conditions. The significant improvement of the enzyme’s stability over time and under harsh reaction conditions can be explained mainly by the creation of stable enzyme–support interactions, which stabilize and stiffen the entire enzyme structure and protect it against thermal and chemical inactivation [65,66]. Moreover, the support material additionally protects laccase against inactivation [67]. These observations are confirmed by the values of the inactivation constant, which for immobilized laccase was found to be 0.0038 1/min, more than four times lower than the value for free laccase. Nevertheless, a drop in the catalytic properties over time and after incubation might be explained by both partial inactivation of the enzyme and its partial elution from the support. Similar observations were made by Yang et al. who immobilized laccase from *Cerrena* sp. by adsorption using chitosan beads as a support. In their study, due to stabilization of the enzyme upon immobilization, the half-life of the immobilized laccase was found to be almost twice that of the free biocatalyst [68]. By contrast, in a study by Tavares et al. in 2015 [69], commercial laccase was immobilized by adsorption using multi-walled carbon nanotubes. Although the immobilized enzyme retained high catalytic properties, surprisingly, its stability decreased upon immobilization.

### 4.2. Removal of Tetracycline

After the effective immobilization of laccase had been confirmed and the chitin-based biocatalytic systems thoroughly characterized in terms of their catalytic properties, the immobilized enzymes were applied in the removal of tetracycline from water solutions under various process conditions, including different pH, temperature, and initial concentration of tetracycline solution. Due to the excellent sorption properties of the chitinous scaffolds, removal by both adsorption and catalytic conversion was investigated in order to evaluate the contribution of each of these pathways to the total removal of the antibiotic. However, it should be noted that upon laccase immobilization, some of the active centers capable of adsorbing tetracycline might be saturated by the biomolecules. Therefore, data on the efficiency of adsorption of tetracycline refer to adsorption of the antibiotic by chitinous scaffolds with thermally inactivated enzyme, and total removal efficiency is based on simultaneous catalytic conversion and adsorption of tetracycline.

The concentration of tetracycline, one of the most frequently used antibiotics, in wastewater varies depending on the source, but it usually ranges from 100 to as much as 1000 μg/L [70]. Thus, in the first stage of the study, the effect of various initial concentrations of tetracycline solution on its removal rate was investigated. The data indicate the clear trend that the higher the concentration of the tetracycline solution is, the lower the percentage of adsorption and the higher the percentage of catalytic conversion in the removal of the antibiotic. This may be explained by the fact that with an increasing amount of tetracycline molecules in the solution, the maximal sorption capacity of the scaffold was attained and further adsorption was extremely limited [17]. The lowest efficiency of removal of tetracycline, around 80%, was recorded for antibiotic solution with a concentration of 3 mg/L. This may be because the amount of pollutant in the solution is too high to allow its removal by adsorption or catalytic action, as previously reported by Yu et al. [71] and Kumar and Cabana [72]. In another study, by Ji et al. (2016), laccase was covalently immobilized on TiO_2_ nanoparticles and used for the removal of pharmaceuticals. Although over 90% of the pollutants were removed from a solution at concentration 5 mg/L, addition of a mediator agent such as syringaldehyde was required to achieve high removal efficiency [73]. Nevertheless, it should be emphasized that the biocatalytic systems produced in this study enable the total removal of tetracycline by simultaneous adsorption and catalytic conversion from solutions at concentrations ranging from 0.1 to 1.0 mg/L.

It is also known that pH might significantly affect both adsorption and catalytic action; therefore, the effect of this parameter on the efficiency of removal of tetracycline was studied over a wide pH range from 3 to 9. Over 80% of tetracycline was removed by simultaneous adsorption and catalytic conversion in the pH range from 3 to 7. However, the data indicate that due to the high sorption capacity of the chitinous scaffolds, the adsorption efficiency was around 50% over the pH range 3–7. Meanwhile, the efficiency of catalytic conversion increased up to pH 5, reached a maximum (40%) at pH 5 and 6, and then decreased. This is related to the fact that in strongly acidic conditions, below pH 4, tetracycline molecules and chitinous scaffolds are negatively charged, leading to ionic repulsions and lowering adsorption efficiency [74]. The lower catalytic conversion rate at basic pH is related to the negative effect of OH^−^ ions, which influence the enzyme’s microenvironment and reduce its activity [75]. Nevertheless, in the pH range 3–6 over 40% of the tetracycline was removed by catalytic conversion, which indicates that the stability of the enzyme was improved upon immobilization, due to the formation of stable enzyme–support interactions and the protective effect of the support [76,77]. Similar observations were made by Shao et al. (2019), who observed the high activity of laccase immobilized by covalent binding onto hollow mesoporous carbon spheres, at pH ranging from 2.5 to 6.5 [78]. By contrast, a significant effect of pH on the removal of hazardous pollutants was observed by Dai et al. (2016) who immobilized laccase using electrospun fibrous membrane modified by multi-walled carbon nanotubes. They reported that over 60% of bisphenol A was removed only in the pH range 3–6 [79].

The effect of temperature on the efficiency of removal of tetracycline was also examined, as temperature can significantly affect both of the processes involved in the removal of the antibiotic. Over the whole analyzed range, the percentage profiles of adsorption and catalytic conversion showed similar trends, although the percentage for adsorption was about 20% higher than for catalytic conversion, this being directly related to the sorption properties of the scaffolds [26] and temperature conditions [80]. The temperature profile of the catalytic conversion of tetracycline by immobilized laccase is directly related to the properties of the laccase from *Trametes versicolor*, which exhibits the best catalytic properties at 25–35 °C [81]. Similar observations were reported by Zhang et al. (2020) [82]. However, as compared to free laccase, the immobilized enzyme exhibited high activity over a wider temperature range, which indicates stabilization of the enzyme structure and its protection against thermal inactivation due to high temperature [83]. Furthermore, it should be highlighted that due to simultaneous adsorption and catalytic conversion, around 80% of the tetracycline was removed over a wide temperature range from 15 to 65 °C. By contrast, in our previous study concerning the immobilization of laccase using electrospun fabricated membranes, the highest removal rate of tetracycline (around 90%) was achieved in a slightly narrower temperature range, from 25 to 45 °C [84].

To sum up briefly, the efficiency of removal of the antibiotic by simultaneous adsorption and catalytic conversion was above 90% over the pH range from 4 to 6 and the temperature range from 25 to 45 °C, from solutions at concentrations up to 1 mg/L. Since the preliminary tests of the use of pure chitinous scaffolds and free enzyme for adsorption and catalytic conversion of tetracycline produced inadequate results (efficiencies of around 80% and 60%, respectively), the data presented suggest that the produced biocatalytic systems might be considered as a sustainable alternative for the removal of pollutants from wastewaters.

Nevertheless, from the practical point of view, one of the most crucial properties is the recyclability of the chitinous scaffolds with immobilized enzyme over repeated removal cycles. The results show that although the efficiency of removal of tetracycline declined slightly with repeated use, even in the 10th batch removal cycle over 90% of the antibiotic was removed. The exceptional reusability of these biocatalytic systems is related to the stability and durability of the chitinous scaffolds [26] as well as the ability of this material to protect the immobilized enzyme against inactivation as a result of process conditions [85]. Furthermore, the enzyme–support interactions protect the enzyme against elution from the matrix, as tetracycline is washed out from the scaffolds between repeated uses [86]. Although the data show that the immobilized laccase is stable during repeated use, the decrease in tetracycline removal is related mainly to inactivation of the enzyme due to its inhibition by the macromolecular products of the reaction [87], although partial elution of the laccase from the support should not be excluded. In another study, laccase was immobilized using bentonite-derived mesoporous materials and used for the removal of tetracycline in 10 removal cycles [88]. The efficiency of removal of the antibiotic gradually declined over repeated reaction steps, and after five cycles the degradation rate did not exceed 50%, significantly lower than the value obtained in our study.

The final stage of the investigation concerned application of the produced biocatalytic systems for the continuous removal of tetracycline in a packed bed reactor over 24 h. The exceptional removal efficiency over the initial process time is related mainly to the above-mentioned excellent sorption properties of the chitinous scaffolds. However, due to saturation of the scaffolds’ active sites capable of adsorbing tetracycline, the longer the duration of the removal process, the higher the percentage contribution of catalytic conversion to the removal efficiency. This is related to the improved operational stability of the laccase upon immobilization, as well as the higher resistance of the immobilized enzyme to thermal and chemical inactivation [83]. Nevertheless, the observed slight decrease in the removal rate might be explained by partial elution of the enzyme from the support and its inactivation due to continuous use. In another study, laccase was immobilized using ceramic membranes for the continuous degradation of tetracycline from real wastewaters, and a relatively high removal rate of over 75% was achieved [89].

## 5. Conclusions

In this study, for the first time, 3D chitin scaffolds from the marine demosponge *A. archeri* were used as a support for effective immobilization of laccase. The immobilized enzyme exhibited high activity retention and high substrate affinity. Moreover, the thermal and storage stability of the immobilized enzyme were significantly improved as compared to the free enzyme, indicating the protective effect of the support on the biomolecules. The produced biocatalytic systems were used for removal of tetracycline, and after 1 h of the process at pH 5 and temperatures of 25 and 35 °C, total removal of the antibiotic was achieved from solutions at relatively high concentration, up to 1 mg/L. Moreover, the system consisting of chitinous scaffolds and immobilized laccase displayed good reusability: even after 10 repeated degradation cycles over 90% of the antibiotic was removed. This indicates the potential of the system for practical application in biotechnological processes. Finally, the possible continuous use of immobilized laccase was tested in a packed bed reactor. It was found that after 24 h of the continuous process over 80% of the pharmaceutical was removed. These features indicate the possibility of efficient removal of tetracycline over wide pH and temperature ranges by simultaneous adsorption and catalytic conversion, both over numerous process steps and in continuous use. Furthermore, they may provide economic advantages for the large-scale practical use of the produced biocatalytic systems in wastewater treatment. Nevertheless, further study concerning optimization of the removal process and evaluation of the effect of various inhibitors on the tetracycline removal rate is still required. In addition, in our opinion, future study related to the use of immobilized enzymes for removal of hazardous pollutants should be focused on practical application of the biocatalytic systems in continuous treatment in bioreactors, and on process optimization to achieve the highest possible removal rate.

## Figures and Tables

**Figure 1 biomolecules-10-00646-f001:**
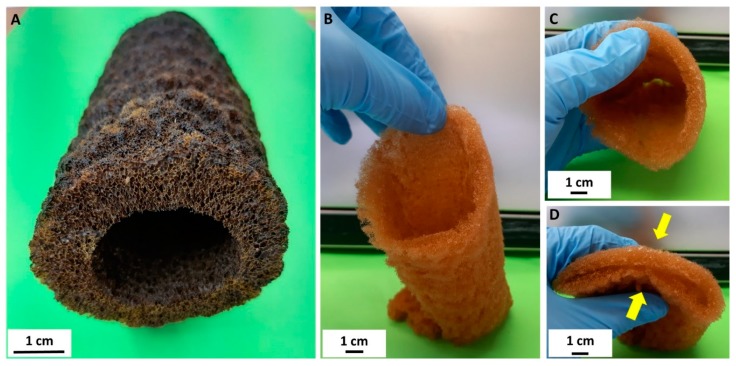
A selected air-dried fragment of *Aplysina archeri* sponge skeleton is rigid and represents the typical tube-like morphology of verongiids (**A**). Decellularization with subsequent demineralization of this construct [6,26] leads to isolation of the flexible chitinous scaffold (**B**–**D**), which reproduces the shape, form, and structural features of the original tubular sponge skeleton, with a length of up to 1.5 m.

**Figure 2 biomolecules-10-00646-f002:**
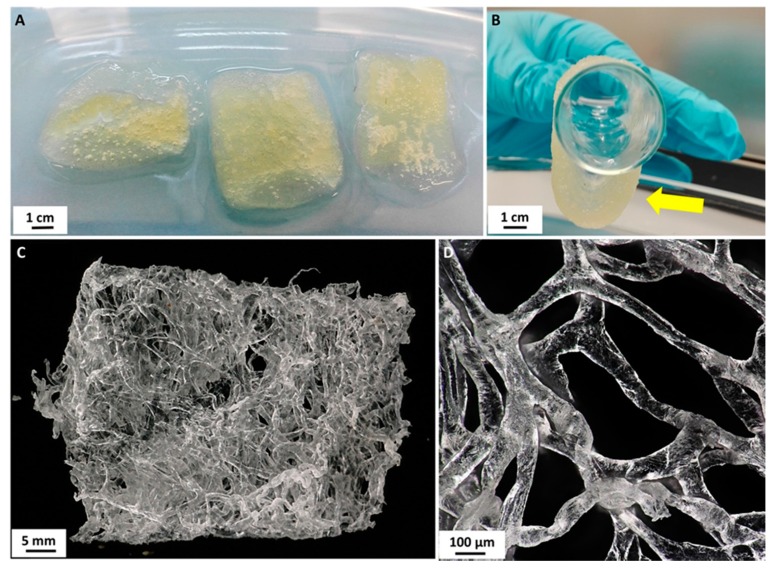
Isolated chitinous scaffolds (**A**) possess excellent ability to insert water (**B**, arrow). On drying in air at ambient temperature these constructs remained three-dimensional (**C**) and retained their characteristic interconnected microtubular architecture (**D**).

**Figure 3 biomolecules-10-00646-f003:**
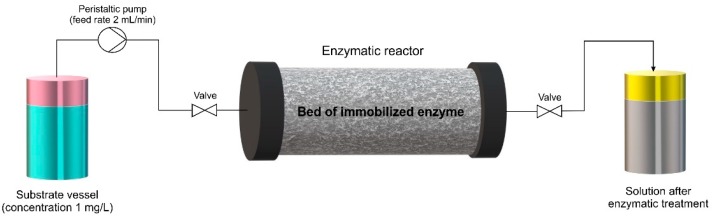
The packed bed reactor system used in the study for continuous removal of tetracycline.

**Figure 4 biomolecules-10-00646-f004:**
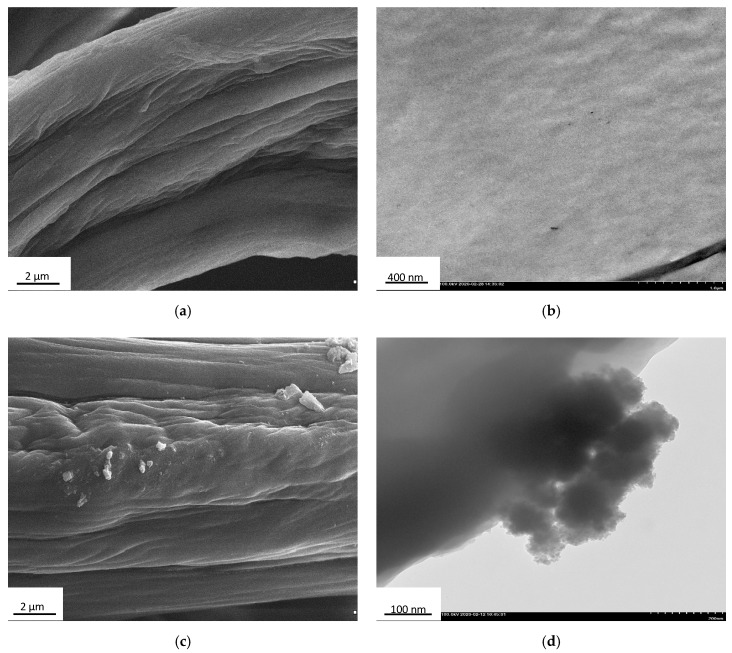
SEM and TEM images of the *A. archeri* sponge skeleton before (**a**,**b**) and after (**c**,**d**) laccase immobilization.

**Figure 5 biomolecules-10-00646-f005:**
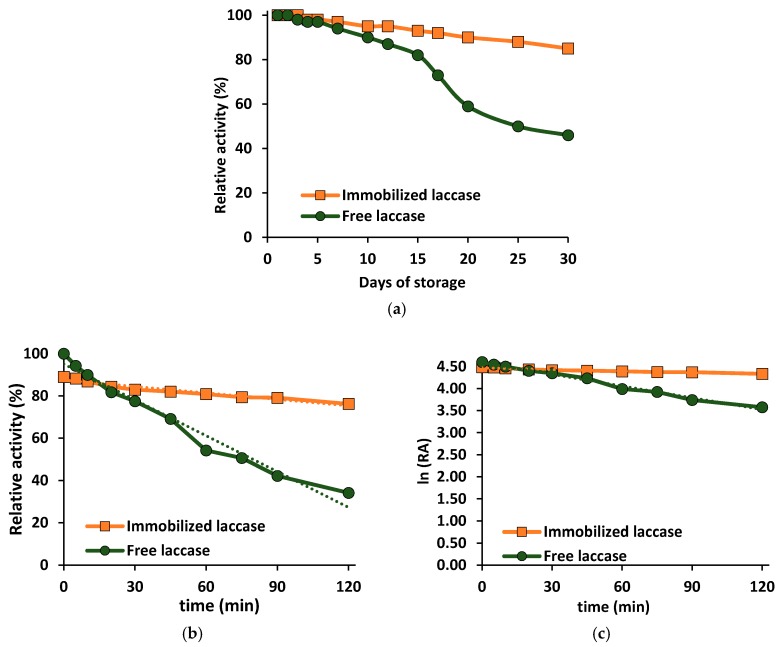
Storage stability of the free and immobilized enzyme (**a**) and stability under optimal process conditions (pH 5, temperature 25 °C) (**b**,**c**). The error value in each of the experiments (based on the mean and standard deviation from three experiments) does not exceed 3.5%.

**Figure 6 biomolecules-10-00646-f006:**
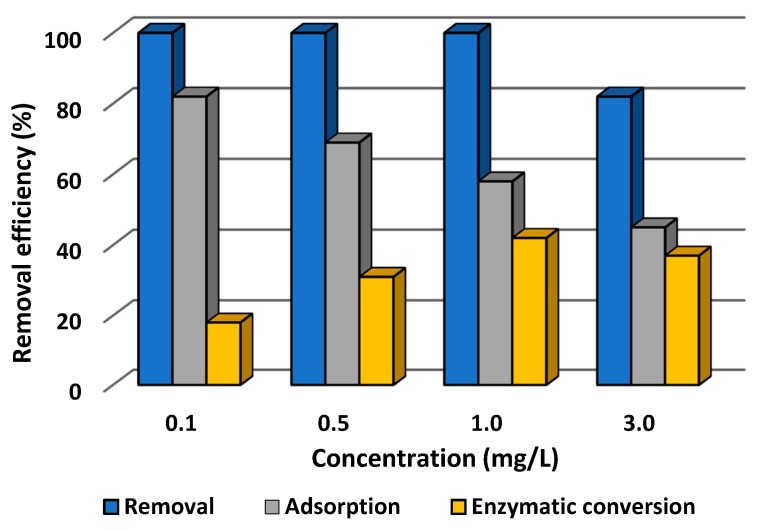
Effect of initial concentration of tetracycline solution on the efficiency of its adsorption, enzymatic conversion, and total removal by laccase immobilized on a chitin scaffold. The error value in each of the experiments (based on the mean and standard deviation from three experiments) does not exceed 3.5%. Adsorption efficiency denotes the efficiency of adsorption by the chitinous scaffold with immobilized laccase following thermal inactivation.

**Figure 7 biomolecules-10-00646-f007:**
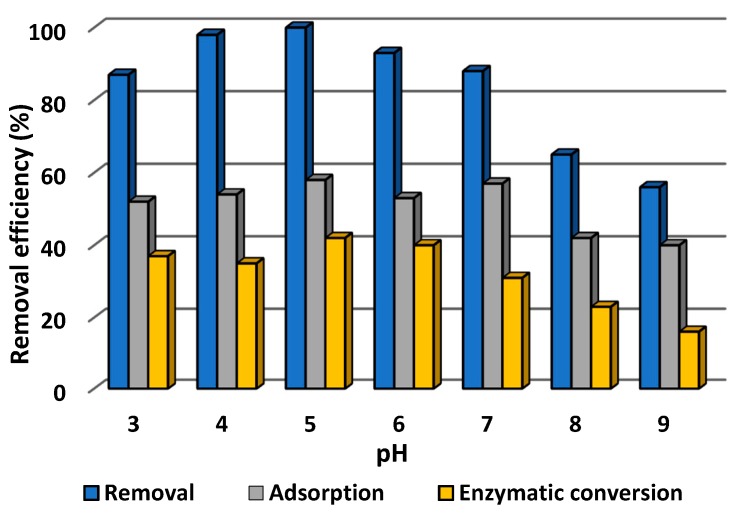
Effect of pH of tetracycline solution on the efficiency of its adsorption, enzymatic conversion, and total removal by laccase immobilized on a chitinous scaffold. The error value in each of the experiments (based on the mean and standard deviation from three experiments) does not exceed 3.5%. Adsorption efficiency denotes the efficiency of adsorption by the chitinous scaffolds with immobilized laccase following thermal inactivation.

**Figure 8 biomolecules-10-00646-f008:**
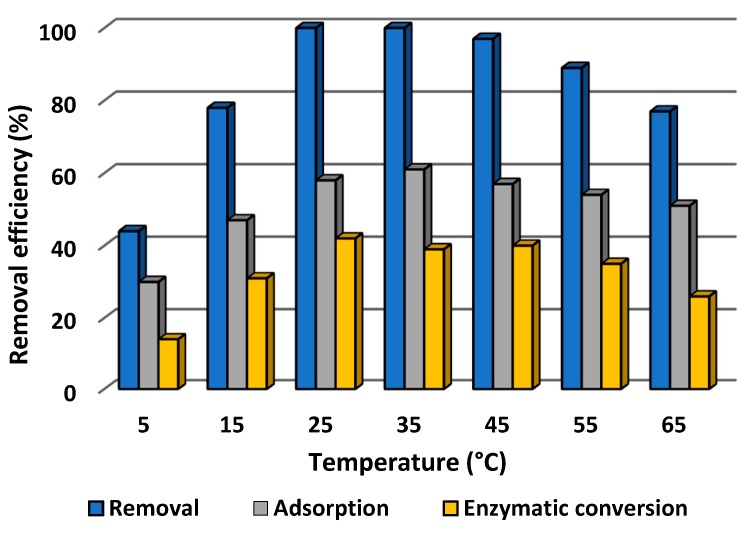
Effect of temperature on the efficiency of adsorption, enzymatic conversion and total removal of tetracycline by laccase immobilized on chitin scaffolds. The error value in each of the experiments (based on the mean and standard deviation from three experiments) does not exceed 3.5%. Adsorption efficiency denotes the efficiency of adsorption by the chitinous scaffolds with immobilized laccase following thermal inactivation.

**Figure 9 biomolecules-10-00646-f009:**
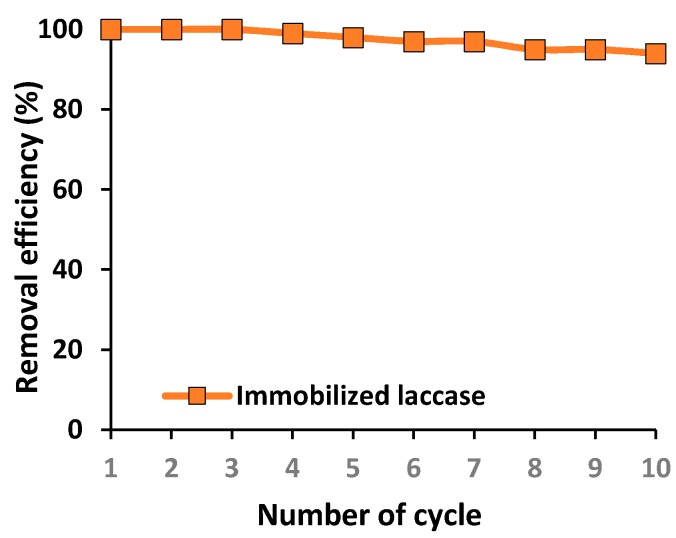
Reusability of the immobilized laccase over repeated catalytic cycles. The error value in each of the experiments (based on the mean and standard deviation from three experiments) does not exceed 3.5%.

**Figure 10 biomolecules-10-00646-f010:**
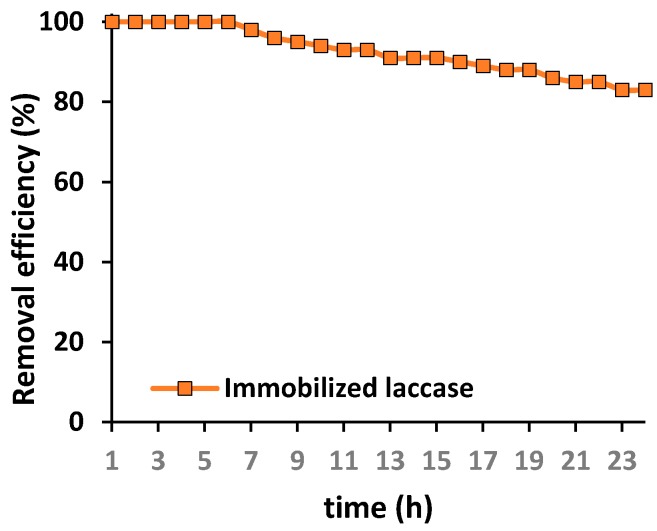
Efficiency of tetracycline removal over 24 h in a packed bed bioreactor containing chitinous scaffolds with immobilized laccase. The error value in each of the experiments (based on the mean and standard deviation from three experiments) does not exceed 3.5%.

**Table 1 biomolecules-10-00646-t001:** Parameters characterizing the immobilization process (amount of the immobilized enzyme, immobilization yield, specific activity and relative activity of free and immobilized laccase, and kinetic parameters of the free and immobilized enzyme.

Analyzed Parameter	Free Laccase	Immobilized Laccase
**Immobilization yield (%)**	-	91 ± 0.6
**Amount of immobilized enzyme (mg/g)**	-	1820 ± 90
**Specific activity (U/mg)**	56 ± 1.9	52.5 ± 1.5
**Activity retention (%)**	-	89 ± 3.2
**K_M_ (mM)**	0.093 ± 0.018	0.113 ± 0.022
**V_max_ (mM/s)**	0.048 ± 0.008	0.045 ± 0.011
**V_max_/K_M_ (1/s)**	0.516	0.398

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
