# Peer review of "3D Chitin Scaffolds from the Marine Demosponge Aplysina archeri as a Support for Laccase Immobilization and Its Use in the Removal of Pharmaceuticals"

_biomolecules, 2020, doi:10.3390/biom10040646_

Round 1

Reviewer 1 Report

The manuscript reports original data on the removal of tetracycline from wastewaters using a laccase immobilized on A chitin 3D scaffold. The structure of the manuscript and experiments are well defined and exposed. The study has been carefully conducted and deals with a topic that should be of interest to readers of Biomolecules. However, some specific matters must be clarified. For these reasons, I recommend the publication of the manuscript as full paper, after minor revision.

  • The calculus of the immobilization yield using the supernatant is not enough. The buffer used to wash the unbound enzyme should be also introduce in the measurement.
  • It is not clear how the authors have measured the activity of the immobilized laccase. Have the authors introduced in the ABTS solution a known amount (weight) of immobilized laccase? The ABTS adsorption to the chitin structure should be also taken in account. Was the immobilization yield determined by dividing the activity value of immobilized laccase, obtained immediately after the immobilization procedure, by the value of activity of the initial laccase solution, converted to U/mg?
  • No information about the used buffers for the pH measurements ranging from 3 to 9 in the removal of tetracycline. In the case of ABTS a monotonic pH profile should appear since ABTS depends exclusively by the OH– inhibition (because the formation of cation radical does not involve proton transfer and the redox potential of ABTS is pH independent). However, the ionic composition of the buffer can also have a critical effect on enzyme activity and therefore, the pH profile should be conducted by the use of a Britton-Robinson universal buffer in order to maintain the same chemical conditions across pH values. Which buffers were used?
  • The agglomerate observed in SEM analysis was identified as laccase clusters however, this is a speculative assumption. An EDX analysis should be performed to confirm the presence of copper in these clusters.

Author Response

Please find the revised manuscript entitled “3D chitin scaffolds from the marine demosponge Aplysina archeri as a support for laccase immobilization and its use in the removal of pharmaceuticals” (Biomolecules-764198), which we hope after revision is now suitable for publication in Biomolecules.

We would like to thank to the Reviewer 1 for insightful review of our work, which contributed to a better understanding of scientific problems relating to the subject of the publication, and will help to eliminate potential errors in the future. Thank also to the Editor for the opportunity to re-submit it, incorporating all of the referee’s suggestions. Our comments and changes are noted below, and are marked in yellow in the manuscript.

Response to Reviewer #1:

The manuscript reports original data on the removal of tetracycline from wastewaters using a laccase immobilized on a chitin 3D scaffold. The structure of the manuscript and experiments are well defined and exposed. The study has been carefully conducted and deals with a topic that should be of interest to readers of Biomolecules. However, some specific matters must be clarified. For these reasons, I recommend the publication of the manuscript as full paper, after minor revision. 

Query 1: The calculus of the immobilization yield using the supernatant is not enough. The buffer used to wash the unbound enzyme should be also introduce in the measurement.

Answer 1: We would like to thank to the Referee for this valuable suggestions. We have checked the amount of the enzyme in the buffer used for removal of bounded enzyme. According to these calculations, we have recalculated the immobilization yield, that was found to be lower and reached 91%. Proper changes have been also provided in the Materials and Methods section and are marked in yellow.

Query 2: It is not clear how the authors have measured the activity of the immobilized laccase. Have the authors introduced in the ABTS solution a known amount (weight) of immobilized laccase? The ABTS adsorption to the chitin structure should be also taken in account. Was the immobilization yield determined by dividing the activity value of immobilized laccase, obtained immediately after the immobilization procedure, by the value of activity of the initial laccase solution, converted to U/mg?

Answer 2: We would like to explain that experiments concerning evaluation of catalytic activity of laccase were performed by addition of 10 mg of free laccase of proper amount of the biocatalytic system that contains 10 mg of immobilized enzyme to 5 mL of 10 mM ABTS solution. The reaction was performed for 60 min and immediately after termination of the process, the absorbance was measured at 420 nm. Based on these data, the activity of the free and immobilized laccase was measured. We would also like to add, that experiment concerning adsorption of ABTS on chitinous scaffolds with thermally inactivated enzyme was performed. Based on the obtained results, it was calculated that less than 10% of the ABTS was adsorbed by the sponge, therefore reduction of ABTS after reaction was considered only as catalytic conversion by free or immobilized enzyme.

The immobilization yield was calculated according to the commonly known and used protocol considering differences in the initial and final amount of the enzyme. However, in this study, according to the Reviewer suggestion, we have also considered amount of the unbounded enzyme washed by the buffer in the calculations. Used approach was in details presented in Section 2.4. Furthermore, to avoid some confusions and for better understanding, this paragraph has been rewritten in the revised version of the manuscript.

Query 3: No information about the used buffers for the pH measurements ranging from 3 to 9 in the removal of tetracycline. In the case of ABTS a monotonic pH profile should appear since ABTS depends exclusively by the OH– inhibition (because the formation of cation radical does not involve proton transfer and the redox potential of ABTS is pH independent). However, the ionic composition of the buffer can also have a critical effect on enzyme activity and therefore, the pH profile should be conducted by the use of a Britton-Robinson universal buffer in order to maintain the same chemical conditions across pH values. Which buffers were used?

Answer 3: Thank to the Reviewer for this comment. We fully agree with the Referee that ABTS in sensitive to the OH- ions, however, tetracycline is significantly less sensitive and more resistant to pH changes. We would like to explain that for the removal of tetracycline commercially available buffer solutions at desired pH were used. At pH from 3 to 6 it was 100 mM acetate buffer as at pH ranging from 7 to 9, 50 mM phosphate buffer was used. It should be emphasized that pH of the buffer solution was not adjusted by the use of HCl or NaOH to obtain desire pH. In all of the experiments buffer solutions at known and measured pH and ionic composition was used. Thus, it might be assumed that composition of the buffer solution do not affect tetracycline structure. Nevertheless, we agree with the Reviewer, that to confirm monolithic pH profiles of the used buffer solutions, their pH profile should be conducted. However, due to COVID-19 pandemic, we are unable to conduct all of the required measurement in a time requested for the revision. The use of the pH buffers at desired pH was also highlighted in the revised version of the manuscript by providing proper changes in the Materials and Methods section (marked in yellow).

Query 4:The agglomerate observed in SEM analysis was identified as laccase clusters however, this is a speculative assumption. An EDX analysis should be performed to confirm the presence of copper in these clusters.

Answer 4: The results of the EDX analysis of the agglomerates observed in SEM photo are presented below. The presence of such elements as carbon, oxygen, nitrogen, sulphur and copper, characteristic for the laccase structure, is clearly seen indicating, that observed agglomerates might be identified as laccase clusters.

Reviewer 2 Report

The manuscript entitled "3D chitin scaffolds from the marine demosponge Aplysina archeri as a support for laccase immobilization and its use in the removal of pharmaceuticals" authored by Zdarta et al., presents the results obtained by immobilization of laccase from Trametes versicolor. The results are interesting and well presented and discussed. The experiments were perfomed in batch and continuous system and the results are very interesting. Prior publication I have some questions and suggestions:

  • The Introduction is focused on the immobilization suport , I would suggest to include some information related to the laccase.
  • The immobilisation as performed at one concentration of laccase solution. How this value was selected?
  • Usually the optimal pH of the enzymes after immobilization suffer a shift to acidic/alkaline values. Why the authors considered pH 5 as optimal pH for native and immobilized enzyme?
  • In the section 3.1.2. the discussion should be improved by comparison the results with other activity/specific activity values previously reported by other groups for the same sources of laccase immobilized by adsorption on different supports.

Author Response

Please find the revised manuscript entitled “3D chitin scaffolds from the marine demosponge Aplysina archeri as a support for laccase immobilization and its use in the removal of pharmaceuticals” (Biomolecules-764198), which we hope after revision is now suitable for publication in Biomolecules.

We would like to thank to the Reviewer 2 for insightful review of our work, which contributed to a better understanding of scientific problems relating to the subject of the publication, and will help to eliminate potential errors in the future. Thank also to the Editor for the opportunity to re-submit it, incorporating all of the referee’s suggestions. Our comments and changes are noted below, and are marked in yellow in the manuscript.

Response to Reviewer #2:

The manuscript entitled "3D chitin scaffolds from the marine demosponge Aplysina archeri as a support for laccase immobilization and its use in the removal of pharmaceuticals" authored by Zdarta et al., presents the results obtained by immobilization of laccase from Trametes versicolor. The results are interesting and well-presented and discussed. The experiments were performed in batch and continuous system and the results are very interesting. Prior publication I have some questions and suggestions:

Query 1: The Introduction is focused on the immobilization suport , I would suggest to include some information related to the laccase.

Answer 1: We would like to thank to the Referee for this suggestion. A paragraph concerning laccase and its catalytic properties was added in the revised version of the manuscript in the Introduction and marked in yellow.

Query 2: The immobilisation as performed at one concentration of laccase solution. How this value was selected?

Answer 2: We would like to explain, that laccase solution at concentration 5 mg/mL was selected as the optimal initial enzyme solution based on results of preliminary experiments (data not presented). In this research initial enzyme solutions at concentration ranging from 0.1 to 10 mg/mL were used. Briefly after immobilization, the specific activity was measured and activity retention was calculated. Obtained data clearly showed that the highest specific activity and relative activity was noticed after immobilization from the solution at concentration 5 mg/mL, therefore, this concentration was selected for further study.

Query 3: Usually the optimal pH of the enzymes after immobilization suffer a shift to acidic/alkaline values. Why the authors considered pH 5 as optimal pH for native and immobilized enzyme?

Answer 3: We fully agree with the Reviewer, that in most cases, upon immobilization, shift in the pH and temperature optima is observed. Nevertheless, we would like to explain that within the frame of the investigation we have performed pH profiles of the free and immobilized enzyme. Based on this data, pH 5 was selected as the optimal, as at this pH conditions both catalysts showed the highest catalytic activity. Furthermore, pH 5 was also found as the optimal for tetracycline removal, as experiment carried out resulted in total removal of antibiotic. We would also like to add that the same pH conditions (pH 5) for both, free and immobilized enzyme (pH shift was not observed upon immobilization) might be related to the adsorption character of the formed enzyme-support interactions. This type of immobilized results in very limited interference of the enzyme structure and in consequence retention of the initial enzyme structure. Moreover, from the obtained data is might be concluded that chitin support material not only provides additional protection for the enzyme molecules but also does not affect significantly microenvironment around enzyme catalytic site. Thanks to the above-mentioned reasons, pH optima was unaltered upon immobilization. 

Query 4: In the section 3.1.2. the discussion should be improved by comparison the results with other activity/specific activity values previously reported by other groups for the same sources of laccase immobilized by adsorption on different supports.

Answer 4: Thank to the Referee for this comment. According to the Reviewers suggestions, we have expanded discussion related to the activity of immobilized enzyme. Another examples of laccase from Trametes versicolor immobilization by adsorption are now presented in Section 4.1 of the manuscript as well as are discussed and compared with our results. Provided changes have been marked in yellow in the revised manuscript.